# Optimizing Eurasian Perch Production: Innovative Aquaculture in Earthen Ponds Using RAS and RAMPS—Economic Perspective

**DOI:** 10.3390/ani14213100

**Published:** 2024-10-27

**Authors:** Anna Hakuć-Błażowska, Konrad Turkowski, Tomasz Kajetan Czarkowski, Daniel Żarski, Sławomir Krejszeff, Jarosław Król, Krzysztof Kupren

**Affiliations:** 1Department of Tourism, Recreation and Ecology, Institute of Engineering and Environmental Protection, Faculty of Geoengineering, University of Warmia and Mazury in Olsztyn, ul. Oczapowskiego 5, 10-719 Olsztyn, Poland; hakuc.blazowska@uwm.edu.pl; 2National Inland Fisheries Research Institute, ul. Oczapowskiego 10, 10-719 Olsztyn, Poland; kturkowski@infish.com.pl (K.T.); t.czarkowski@infish.com.pl (T.K.C.); s.krejszeff@infish.com.pl (S.K.); j.krol@infish.com.pl (J.K.); 3Department of Gamete and Embryo Biology, Institute of Animal Reproduction and Food Research of Polish Academy of Sciences, ul. Tuwima 10, 10-748 Olsztyn, Poland; d.zarski@pan.olsztyn.pl

**Keywords:** rural areas, fish farming diversification, earth ponds, RAS, RAMPS, production profitability, Eurasian perch, aquaculture

## Abstract

The conceptual assumptions and economic aspects of an innovative and field-proven method for the production of consumer-attractive Eurasian perch (*Perca fluviatilis* L.) based on rearing in recirculating aquaculture systems and adapted earthen ponds for the production of common carp (*Cyprinus carpio* L.) are presented. The key to success was the optimization of post-season reproduction and rearing procedures for larvae of this species and the production of this predatory fish at high densities in earthen ponds to consumable sizes. It is shown that, despite the high variable costs (mainly labor and feed), this type of production can be a promising option for achieving food security and economic development in some EU countries. The article also examines the potential opportunities and risks associated with fish production in recirculating aquaculture systems and recirculating aquaculture multitrophic pond systems.

## 1. Introduction

The main objective of the EU Common Fisheries Policy (CFP) 2021–2027 is to promote an innovative and sustainable Blue Economy in the fisheries and aquaculture sector and to meet commitments to global processes for responsible environmental protection and sustainable use of aquatic resources. In 2016, the EU Blue Economy was estimated to have EUR 174.2 billion in gross value added and 3.48 million jobs. For the Blue Economy, the CFP aims to enable the growth of a sustainable blue economy and to support communities in areas dependent on fisheries and aquaculture. An important condition for stimulating such growth is the need for innovative efforts to develop efficient production methods for “new species” and to increase the technological level of the entire sector [1,2]. It is worth noting that diversification based on native species, which are highly valued by consumers and offer, among other things, the possibility of exploiting the production potential of earthen ponds and increasing the production revenues and profitability of carp farms, is a priority for the development of the freshwater aquaculture sector throughout the European Union.

The Eurasian perch (*Perca fluviatilis* L.) is a very important species for both commercial and recreational fisheries [3,4,5,6,7,8,9,10,11,12]; the intensive culture of this species is still in its infancy, and market demand for this species is largely unsatisfied [13,14]. The culture of perch species, including Eurasian perch, is perfectly in line with the development of the EU aquaculture industry. The development of the farming of this group of fish is supported both by consumer expectations for healthy, high-quality animal protein and by the reduction of open-water fishing, which is the main source of demand for products derived from these fish species. Currently, the culture of all perch fish in the world accounts for only about 1% of total aquaculture production, although new farms using recirculating aquaculture systems are steadily emerging in countries such as Switzerland, Ireland, France, and Denmark. Catches of Eurasian perch in surface waters have been stable for about 30 years, reaching nearly 32,000 metric tons per year in 2019. In contrast, global aquaculture production of Eurasian perch has been on an upward trend, reaching only 182 metric tons in 2011 [15] and 500 metric tons for the first time in 2013 [16]. From limited sources, it is difficult to draw detailed conclusions on the dynamics of this production, but from a study by Fontaine and Teletchea [14], for example, we learn that in France, there are three perch farms with a total production of 100 tons per year.

### 1.1. Characteristics of Earthen Pond Aquaculture

The technology of fish production in earthen ponds, where relatively low water flow is maintained, is mainly based on the traditional extensive rearing of monoculture carp (*Cyprinus carpio* L.). Until the late 1970s, this technology was based on stocking material obtained by natural spawning in small, shallow ponds. In later stages of rearing, other types of earthen ponds (e.g., larval, juvenile) were used, adapted to the various and changing requirements and sizes of the fish [17]. Traditional carp-rearing technology is limited by low efficiency in the production of stocking material due to the low possibility of controlling natural spawning and the dependence of this stage of production on weather conditions. Therefore, in the last four decades, more and more farms in Europe, as well as in Poland, have started to implement controlled reproduction with hormonal simulation [18]. This has allowed, among other things, greater control over the fish reproduction process and greater flexibility in planning the timing of spawning. In many cases, this has led to a reduction in production costs and, as a result, an increase in the competitiveness of such farms in the market. Many farms have unlocked their production potential, both in terms of land-based spawning ponds and fry ponds, which are often not currently used in the production cycle. This has paved the way for the implementation of the Recirculating Aquaculture Multitrophic Pond System (RAMPS). Meanwhile, further advances in carp production technology, such as the use of the Recirculating Aquaculture System (RAS), potentially offer even more opportunities to improve the efficiency of fish producers through the production of additional fish (including highly valued predatory fish).

The estimated production of additional fish in carp ponds is currently at a low level and, for example, in Poland, does not exceed 10%, i.e., about 1.7 thousand tons [19] of the total production obtained from extensive aquaculture systems. Among these fish, a small percentage are predatory species (pike (*Esox Lucius* L.), catfish (*Siluris glanis* L.), pikeperch (*Sander lucioperca* L.), and Eurasian perch), mainly due to their different environmental and food preferences compared to common carp, to whose needs these systems are adapted. At the same time, the production of predatory fish in monoculture in earthen ponds has not yet been tested on a large scale, especially with the use of compound feeds, which guarantee high growth rates and higher potential profitability of this type of intensified technology.

### 1.2. Percid Fish Aquaculture

Among predatory fishes, percid fishes (perch, pikeperch) are among the most highly valued in several European countries and achieve high market prices (in Poland and abroad). In recent years, these two species have become the most frequently identified taxa for the diversification of freshwater aquaculture [20]. However, it is important to note that the final/actual list of most attractive species may differ slightly in different European countries due to consumer preferences or different types of aquaculture facilities. It is worth noting here that the diversification of production in inland waters, based on species highly valued by consumers, is one of the priorities for the development of the aquaculture sector in the European Union [21]. In the case of percid fish culture, the most effective method is production in the RAS, taking into account high (22–25 °C) constant temperature, appropriate dissolved oxygen in the water, and feeding the fish with high-quality compound feed. Under such conditions, commercial fish can be obtained in less than one year (in the case of perch) or less than two years (in the case of pikeperch) [20,22]. However, RAS production technology, despite its great prospects, is still a method that producers approach with great uncertainty. This is due to the high initial investment costs, the long amortization period (the first revenues are only expected after 5–6 years), the need to employ qualified personnel, and the high production costs (mainly feed, labor, and electricity) [23,24,25].

The above arguments allow us to conclude that this type of technology is developed mainly in highly industrialized countries (e.g., France, Germany, Benelux, Ireland, Denmark, or Italy) [15,20]. Currently, in Poland, there are several companies that have started breeding perch in the RAS; however, to our knowledge, none of them maintains a constant level of production, and there are no data on the basic indicators that would allow a reliable estimate of the economic profitability of an ego type of venture. Current aquaculture production of these species is mainly targeted at niche premium markets [26], with limited scale-up of production [27]. There appears to be considerable potential for growth in this sector, given the rate of expansion and the average price, which is currently close to that of salmonids (fillet at EUR 10–12) [28]. However, as with aquaculture production in general, price pressure from consumers is forcing research and development efforts to focus on reducing production costs [20].

Assessing the progress made in each sector since the last major review of the scientific literature was published in 2019, it is clear that European perch aquaculture is moving away from its status as a candidate to become an established European freshwater aquaculture industry. However, there are still significant bottlenecks that need to be addressed, from spawning stock management to production cost optimization [20].

### 1.3. Overview of RAS and RAMPS Combined Technology

The production of percid species in the RAS has been practiced in Europe for almost 20 years. However, the technology requires continuous optimization, such as increasing the efficiency of juvenile rearing, which currently fluctuates around 10%. This is due to the lack of standardized procedures for stimulated reproduction and larval rearing, which are subject to constant modification. In addition, laboratory-scale experience is not easily transferable to industrial scale. Therefore, the production of perch fry in the RAS, although currently feasible, is still characterized by major constraints related to [14,29] as follows:Low or highly variable reproductive efficiency;Low efficiency of larval rearing (lack of standardized procedures);Lack of rearing systems dedicated to perch;Lack of implementation of monosexual and triploid breeding as a result of genomic manipulation;Highly imperfect breeding programs that should ensure the maintenance of economically desirable traits over generations;Lack of production-proven disease prevention and treatment measures and pathogen monitoring, which are particularly dangerous at high stocking densities;Differences in growth rates related to species adaptation during life in the wild;Consideration of factors that cause juvenile mortality, including larval aberrations and fungal diseases;Damage compensation measures and protection from predators, e.g., cormorants *Phalacrocorax carbo* L. 1758 or Eurasian otter *Lutra lutra*.

Factors limiting the dynamic development of the RAS for percid species have forced the search for alternative methods to reduce production costs. For this reason, a perch production technology based on the production of stocking material in the RAS and the subsequent transfer of these fish to earthen ponds for the production of commercial-size fish has been implemented in Ireland. This concept is based on many years of experience, which clearly shows the following [24,30]:Controlled reproduction of fish is the most efficient method of producing high-quality eggs;RAS technology allows for off-season reproduction at any time of the year;The control of the reproductive process makes it possible to selectively cross fish and thus conduct long-term breeding programs;The rearing of larvae and juveniles in the RAS is the most effective way to obtain high-quality stocking material (with a unit weight of about 5 g) despite potentially high costs;The cost of rearing broodstock in the RAS is very high and requires the construction of large aquaculture facilities, while the infrastructure for rearing juveniles typically represents 15–30% of the total investment cost;The ability to rear fish in earthen ponds potentially reduces the cost of raising commercial fish by about 60–70%.

The above elements supported the development of a technology based on the integration of two production systems—the rearing of broodstock in the RAS and the production of commercial-size fish in earthen ponds.

To date, there is a lack of published data on the monoculture production of Eurasian perch in earthen ponds. The only available results come from Ireland (Damien Toner, BIM, Ireland—oral information), where a number of trials have been carried out in recent years to investigate the possibility of rearing perch. These experiments have led to the development of the so-called “pill pond” concept, which takes into account the need for fish density (at both poles of the pond) while the rest of the facility is used only for water cleaning. In such a system, the water movement is additionally accelerated by a paddle wheel driven by an efficient electric motor with low power consumption [31,32,33].

The essence of this production system is the need for fish stocking densities to efficiently feed the fish with commercial feeds. Stocking density is a factor that allows for efficient feed distribution while maintaining good water quality provided by forced water flow. Advantages of this system include the following:Low energy costs through the ability to integrate the system with alternative energy sources (e.g., photovoltaic panels, windmills, etc.);Minimization of employment through the use of automated feeding;Production efficiency under Irish conditions of up to 3–4 tons per hectare of pond (taking into account the entire pond area, with the “non-production” part, i.e., the water treatment zone);A production cycle of commercial fish in less than 12 months, with integration with the production period of stocking material (fry) in the RAS.

An important aspect of the whole issue is that the Irish production system discussed was based on the idea of building new facilities on wasteland (e.g., post-mining peatlands), where the primary productivity of ponds does not play a major role. Contrary to Ireland, Poland has a huge potential to transform the existing pond infrastructure at a relatively low cost. Of course, this will require an innovative approach, different from the Irish model, and a series of tests on a commercial scale to refine the ability to use this type of fish production technology in our country. However, even before the PRO-PERCH project [34] began, a number of pilot studies conducted in Hungary and the Czech Republic indicated that, despite the climatic differences, the efficiency of rearing perch in earthen ponds should be comparable to, and perhaps superior to, that recorded in Ireland.

Following theoretical considerations, consultations with producers, and conceptual modeling, the following technological assumptions relevant to Eurasian perch production in the RAS and RAMPS were extracted:Controlled spawning of Eurasian perch will take place outside the breeding season (in January);The rearing of fry in the recirculation system will last no longer than 3 months (until the end of April at the latest);Semi-intensive fattening in earthen ponds will be carried out in monoculture with feeding of fish with compound feed;The adaptation of the earthen ponds of carp type (spawning ponds, fry ponds, fingerlings ponds, storage ponds) to the requirements of this technology should take into account their peculiarities and be characterized by the minimum cost of implementation of the project (Figure 1 and Figure 2).

Taking the above into account, the preliminary assumptions of integrated Eurasian perch production technology using the RAS and RAMPS are shown in Figure 2.

One of the key reasons that determines whether it makes sense to put a particular project into practice is its profitability. Assessing profitability requires conducting cost analysis and revenue analysis, estimating income or profit, and establishing threshold (critical) conditions for the economic efficiency of the venture [35]. Of course, it should not be forgotten that in addition to production costs, the economic feasibility of the project also depends on market conditions of supply and demand [36].

The purpose of this study, in addition to presenting its conceptual framework, was to analyze the costs and profitability of Eurasian perch production through a novel method using the RAS and RAMPS. Given the lack of detailed information on the subject, the aim of this study was to present the conceptual and technical framework as well as to analyze the costs and profitability of producing Eurasian perch using an innovative method, lasting approximately one year, using the RAS and RAMPS. Our motivation is that the production data presented in this manuscript, although obtained at the semi-production scale, should contribute to the discussion on improving the overall efficiency of Eurasian perch production, which, according to the available literature, is still one of the major obstacles limiting further expansion of the sector [20].

## 2. Materials and Methods

In order to carry out an economic analysis of the production of Eurasian perch broodstock under controlled conditions, the data obtained from the activities carried out within the PRO-PERCH project and presented in Table 1 were used. The economic analyses were based on the actual costs of the activities carried out in Polish zlotys (PLN) at the current prices in 2022 (EUR 1 = PLN 4.69). The production cycle of perch fry was carried out in three stages.

The entire process of producing Eurasian perch fry was carried out using the existing hatchery equipment used for carp production. This type of equipment is now a standard and integral part of most earthen pond carp farms. The rearing of Eurasian perch fry in the three systems described above (three production stages—larviculture period) took a total of 177 days (Figure 1). The first stage, related to the preparation of spawners, took 75 days. The second stage, related to the incubation of eggs and rearing of larvae, took another 57 days. The third stage, related to the rearing of fry, took 45 days.

For the economic analysis of the intensive production of Eurasian perch in earthen ponds (184 days), the direct costs of rearing were taken into account based on the assumption that the separation and adaptation of a production area of 0.2 ha (two ponds, e.g., fish storage facilities of 0.1 ha each) from the existing earthen ponds on the farm for rearing commercial perch will not lead to changes in indirect costs, such as administrative costs. Similar principles were applied in other studies of the economic effectiveness of such ventures [35,37]. The rearing costs were analyzed by type and divided into fixed and variable costs. Variable costs depended on stocking density and the length of the rearing period; however, fixed costs comprised of depreciation and capital costs were unchanged. Costs and profits were analyzed, and then the threshold parameters for pond production of fish of commercial size, which guarantee the profitability of farming, were calculated as follows: break-even point; minimum price. The economic safety margin of rearing was calculated based on the price of commercial fish for consumption and based on the level of their production. These parameters are elements of the so-called profitability threshold analysis [38,39].

The rules for calculating the various parameters of the economic efficiency analysis are presented and discussed in detail in an article on the economic aspects of larviculture of three species of rheophilic fish [35]. The formulas used for the calculations of the European perch culture are presented together with detailed results in the following chapters of this paper.

## 3. Results

### 3.1. Production Costs of Eurasian Perch Fry (with an Average Weight of 5 g) in the RAS

The induced reproduction and initial rearing resulted in an initial stocking of 86,400 freshly hatched fish and a final stocking of 43,200 fry with an average weight of 0.5 g each. In the third stage, which lasted 45 days, from a stocking of 43,200 0.5 g fry, 36,288 fry/fingerlings were produced with an average weight of 5 g each. The survival rate during this period was 84%. The total direct costs recorded during the above stages are summarized in Table 2.

The cost of producing Eurasian perch larvae was dominated by labor costs (41%), followed by feed (food and feed) (25%), and electricity (23%). Other cost components were at a low level of 3–4% (Table 3).

In the case of feed (25% of the total cost), the cost of the first feed, i.e., Artemia nauplius (34%), was the highest. The total cost of feeding at this stage amounted to PLN 2695.55, which was 58% of the total cost of feeding during the whole production cycle of perch stock (Table 2 and Table 3). At a similar level were the costs of rearing fry (41%), while the costs of feeding spawners were relatively low (1%) (Table 2).

The highest labor input (41% of the total cost) was characterized by the egg incubation stage (53%), followed by the holding of spawners (34%), while the fry-rearing stage (14%) required a much lower labor input (Table 2 and Table 3).

Another important component of Eurasian perch production costs was electricity (23% of total perch fry production costs). Electricity consumption was more or less evenly distributed among the three stages of perch fry production. It was highest during fry rearing (39%), followed by larval incubation and pre-rearing (32%), and slightly lower during spawner rearing (29%) (Table 2 and Table 3).

As already mentioned, the share of the other cost components, water and wastewater consumption, disinfectants and detergents, and the means necessary for the handling of spawners, was low, at 3–4% in each cost case (Table 2).

Considering the 36 288 produced perch fry/fingerlings weighing 5 g and the total cost of production amounting to PLN 18,062.00 (Table 2), the unit cost of fry/fingerling production was PLN 0.50 per fry.

### 3.2. Economic Analysis of the Intensive Production of the Eurasian Perch in Earthen Ponds

In the calculations, the costs of supervision (PLN 4609.80) and depreciation (PLN 5682.00) were considered fixed costs (PLN 10,292.00), while the remaining costs (PLN 48,158.00) were considered variable costs of rearing (Table 4). The depreciation costs were calculated on the basis of the investment costs for the purchase and installation of the equipment for the production of fish of commercial size (Table 4).

Depreciation was calculated using the straight-line method. The basis was the expected 20-year life of the equipment and the amount of capital expenditure incurred (Table 5).

Costs were dominated by the purchase of feed (45%) and stocking material (25%). Other cost components included electricity (11%), labor (10%), and depreciation of equipment (10%). The cost of stocking material (PLN 14,667.00) was the purchase of 29,334 perch fry with an average weight of 0.005 kg each (total weight 146.67 kg). Labor costs were mainly for supervision (88%). The remaining 12% of the costs were labor costs related to stocking (6%) and catching fish in the pond (6%) (Table 4).

### 3.3. Revenues and Profit

As a result of rearing fish in earthen ponds, 2640.06 Eurasian perch were caught with an average weight of 0.10 kg/pc. A total of 2640.06 kg of perch were harvested. This production period had a high survival rate of 90%.

The average price of PLN 25/kg for commercial size perch generated revenue (the total amount of money generated by the activities carried out, measured over a certain period of time, i.e., gross income before deducting any costs):2640.06 kg × 25 PLN/kg = 66,001.50 PLN

The profit was calculated as the revenue minus the cost of raising the fish in the pond (Table 4):66,001.50 PLN − 58,450.00 PLN = 7552.50 PLN

It should be noted that this is the profit generated by the area of 0.2 hectares of specially adapted earthen ponds (two ponds of 0.1 hectares each). Both profit and revenue in this study define potential volumes, i.e., those that will occur if 100% of the Eurasian perch produced is sold. A full analysis of sales opportunities requires appropriate market and marketing studies that are beyond the scope of this study. The assumption of 100% saleability of the production formed the basis for further calculations, including the break-even point and other parameters that determine the conditions for profitability of the rearing.

### 3.4. Break Even Point

The break-even point refers to the volume of production that revenues equal to the costs incurred. Further increases in production, under the same price conditions, generate revenues above costs. Thus, the more production increases above the BEP, the more profitable the venture will be. BEP was calculated according to the following formula:Q = K_s_/(c − a)(1)
where:

Q—break-even point—threshold production volume (kg)K_s_—fixed costs (PLN 10,292.00)c—market price of perch (PLN 25.00/kg)a—variable costs per unit (PLN/kg)

The average variable costs were calculated according to the following formula:a = K_z_/P (2)
where:

a—average variable costs of commercial fish production (PLN/kg)K_z_—variable direct costs (PLN 48,158.00)P—the quantity of (potential) production obtained (2640.06 kg)

a = 48,158.00/2640.06 kg = 18.24 PLN/kg

Q = (10,292.00 PLN)/(25 PLN/kg – 18.24 PLN/kg) = 1522.45 kg

The calculated production volume of 1522.45 kg guarantees a return on costs and no income. Lower production means losses and higher production means profitability is proportional to the increase in production.

### 3.5. Margin of Safety Due to the Level of Production

We determined the percentage difference between the actual production volume that can be achieved in a given technological and cost system and the production volume that defines the break-even point. The safety margin of the enterprise due to the level of production (MOS) is calculated according to the formula:MOS = (P − Q)/P × 100%(3)
where:

MOS—safety margin of the enterprise due to the level of production (%)P—the quantity of (potential) production obtained (2640.06 kg)Q—break-even point—threshold production volume (1522.45 kg)


MOS = (2640.06 – 1522.45)/2640.06·100% = 42.3%


The result obtained (42.3%) indicates that the rearing of Eurasian perch in earthen ponds has a satisfactory safety margin due to the level of production achievable.

### 3.6. Minimum Price

The minimum price is the lowest price at which production is neither loss-making nor profitable, i.e., at which costs are equal to revenues.
c_min_ = (K_s_/P) + a(4)
where:

c_min_—minimum price (PLN/kg)K_s_—fixed costs (PLN 10,292.00)P—the quantity of (potential) production obtained (2640.06 kg)a—average variable costs of commercial fish production (PLN 18.24/kg)


c_min_ = (10,292.00 PLN)/(2640.06 kg) + 18.24 PLN/kg = 22.14 PLN/kg


The result indicates that the sale of perch below PLN 22.14/kg will be loss-making.

### 3.7. Margin of Safety of the Venture Due to Price

The margins of safety of the venture were also analyzed in the study in terms of the price of commercial fish for consumption (MSVP). MSVP was calculated as follows:MSVP = (c − c_min_)/c × 100%(5)
where:

MSVP—margins of safety of the venture due to price (%)c—market price of perch (PLN 25.00/kg)c_min_—minimum price (PLN 22.14/kg)

MSVP = (25.00 PLN/kg – 22.14 PLN/kg)/(25.00 PLN/kg) × 100% = 11.4%

The result (11.4%) indicates that the analyzed venture has a small margin of safety, which allows for a relatively narrow range of price negotiations.

## 4. Discussion

The main part of Eurasian perch production in Europe is the catch of this species in open waters (inland waters, lagoons, and Baltic bays). The smallest and even marginal part is the production of Eurasian perch in aquaculture, where only a few tons are produced annually, including the production of stocking material. The PRO-PERCH project focused on the possibility of using the existing production infrastructure in a more innovative way than before. To date, in many Central European countries, including Poland, the main species produced in earthen ponds is carp. The production of this species contributes significantly to the aquaculture of these countries. In Poland, with a total production of 40,000 tons of fish, about half of it is carp production. In other Central European countries, such as Hungary and the Czech Republic, this share is even higher [40]. It should be noted that this culture is carried out in a very extensive traditional way, which has changed little over several hundred years [17]. Most ponds produce Eurasian perch and zander in addition to carp, but these are usually by-catches, and the production of these species is not planned in advance and is more or less unknown to the farmer. The producers consider perch and zander to be promising species for aquaculture, and 2/3 of them would be interested in more intensive production of this species based on the existing farm infrastructure. The final decision to start production would be strongly influenced by the provision of expert advice both before and during the production of this species, in addition to profitable production.

The above-described novel approach is based on the fact that, increasingly, in European countries where carp are farmed extensively, a large part of the pond area is unused. For example, in Poland, out of about 70,000 hectares of ponds, about 10,000 hectares are currently unused for production purposes. Mainly, this concerns spawning and fry ponds, which make up about 25% of the area on average [41,42]. This is due to the introduction of RAS-based hatchery techniques into standard carp production technology (especially during spawning and fry rearing). Thus, based on the author’s concept, taking into account the modification of the function of these 10,000 hectares of ponds for the production of predatory fish (or other alternative species), it is possible to practically double the production of fish from currently extensive production systems (assuming production in innovative breeding systems of at least 3–4 tons per hectare).

In terms of consumers’ perceptions of Eurasian perch, it is consumed less frequently than other freshwater fish species or bi-environmental fish, which are dominated by Atlantic salmon (*Salmo salar* L.) and Rainbow trout (*Oncorhynchus mykiss*). More than half of the respondents said they did not eat these species at all. The next most common answer was “several times a year” (29.4% of responses). Among those who eat perch, the taste qualities are rated very highly, significantly higher than those of zander. Nevertheless, according to the survey, 13% of Poles would like to eat it more often, preferably in the form of a fillet served in a restaurant. The price of the product available at the point of sale is the most decisive factor for its purchase (survey—unpublished data).

The biggest opportunities for increasing perch consumption seem to be the increasing consumption of fish each year and the fact that customers are increasingly paying attention to origin and would be most likely to consume locally produced fish [43]. Any venture that changes or diversifies an existing business must be profitable from the producer’s point of view and acceptable from the consumer’s purchasing capacity. As part of the Pro-perch project, experiments were carried out under Polish conditions, and an analysis of the profitability of the venture was carried out for this area on the basis of the real costs incurred.

The total direct cost of rearing 29,334 perch fry (0.005 kg each) in 0.2 ha of earthen ponds for six months was PLN 58,450.00. As a result, 2640.06 kg of perch with an average weight of 0.1 kg (consumption size) were harvested, and the profit amounted to PLN 7552.00. The production per hectare of pond area would be about 12,5 tons of perch, and the expected profit would be PLN 37,500.00. The average production yield of common carp in Poland is 277 kg/ha [44]. It should be noted that the fry-rearing systems used in the analysis allowed the production of a surplus of almost 7000 fry weighing 5 g, which exceeds the rearing needs of two ponds of 0.1 ha each. Adding the additional funds from the sale of this stock, and only at the cost of production, would increase the income by almost half. In both parts of the experiment (RAS and earthen ponds), variable costs were the largest part of the total costs. In the case of fry rearing, these were the costs of labor (41%), feed (26%), and electricity (23%). At the stage of rearing perch in earthen ponds, feed costs dominated (45%), and other costs such as labor, electricity, and depreciation of equipment did not exceed 30% of the costs for this case total.

The cost structure of Eurasian perch production presented above is similar to previously analyzed cases of rearing other fish species in the RAS or earthen ponds [35,45,46,47,48,49]. The difference in the level and structure of the costs of rearing in the RAS versus earthen ponds is mainly based on labor and feed costs. For the production of other species, such as catfish (*Siluris glanis* L.), trout (*Oncorhynchus mykiss*), and bass (*Dicentrarchus labrax* L.), feed, labor, and energy costs were identified as major contributors to total production costs, but the order of importance varied by species/system/scale. Improvements in feed conversion and labor efficiency have the greatest potential to reduce unit production costs and may be as important as the scale of production in achieving profitability [45]. Under Polish conditions, the cost structure is similar both for carp (*Cyprinus carpio* L.) production in earthen ponds [48] and for asp (*Aspius aspius* L.), ide (*Leuciscus idus* L.), or dace (*Leuciscus leuciscus* L.) production in the RAS [35,49]. It should be noted that RAS-based systems are currently more expensive than other forms of production and are not always profitable. Capital, energy, and labor costs are higher for the RAS than for other production systems, and industry growth will require cost reductions in these areas [45,50]. This is particularly important in light of recent increases in electricity costs and wages. Cost reduction and, thus, the development of RAS-based livestock farming are increasingly promoted by a variety of innovative technologies using alternative energy sources.

The production system described in this paper, combining the RAS for the production of Eurasian perch fry and RAMPS for the rearing of fish for consumption, allowed the venture to become profitable. An important element in the profitability of this venture was the use of existing production infrastructure. The rearing of perch for consumption was characterized by a large margin of safety due to the level of production (42.3%) and a small margin of safety of the venture due to the level of prices (11.4%). Further opportunities to improve the profitability of the farm should be sought in reducing the cost of producing stocking material, increasing the scale of production, using equally effective but cheaper feed (including that produced on the farm), and, in particular, selling processed perch (e.g., gutted, vacuum-packed fillets, etc.). Of course, it should not be forgotten that in addition to production costs, the economic feasibility of the project also depends on market conditions of supply and demand as well as the market price of the product [36]. The applied perch production system provides opportunities to reconcile the relatively high environmental requirements of perch with the challenges of reducing production costs. The advantage of the developed production system is that it is easily adaptable to the requirements of other species, thus matching supply with demand. However, it should be noted that the technology for monitoring perch farming and the final list of potential threats under various conditions is still incomplete and needs to be expanded. Its gradual completion will potentially be an important factor in increasing the profitability of production [14,20,29].

## 5. Conclusions

The above conclusions allow us to assume that the trend of aquaculture development towards diversification of activities, especially on the basis of existing infrastructure, has a chance to meet, at least partially, the demands of consumers and producers. The use of a wide range of innovations (technical, market, or institutional) in production based on earthen ponds in carp farms (e.g., new forms of products, renewable energy sources, aquaponics, etc.) is part of this development trend [51]. In the process of diversifying the production function of earthen ponds, the important ecosystem role of these objects should not be forgotten [52,53,54]. Nevertheless, it is important to emphasize the very important role of research and development and technology transfer in this field so that the whole production concept proposed by the PRO-PERCH project can meet the requirements of companies and potential investors.

## Figures and Tables

**Figure 1 animals-14-03100-f001:**
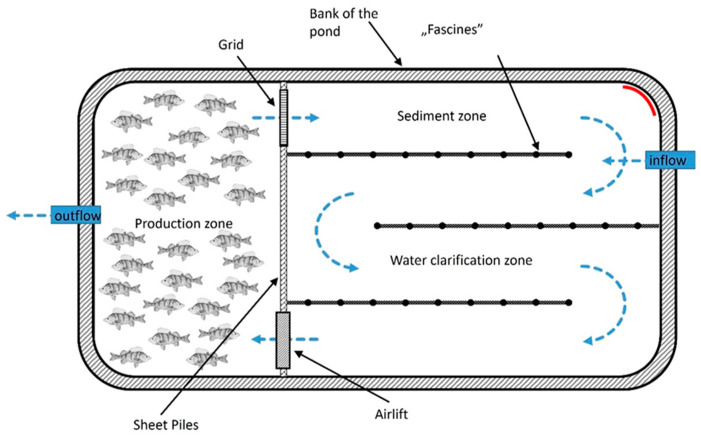
Schematic diagram of the Recirculating Aquaculture Multitrophic Pond System (RAMPS) used for Eurasian perch production. Source: own elaboration.

**Figure 2 animals-14-03100-f002:**
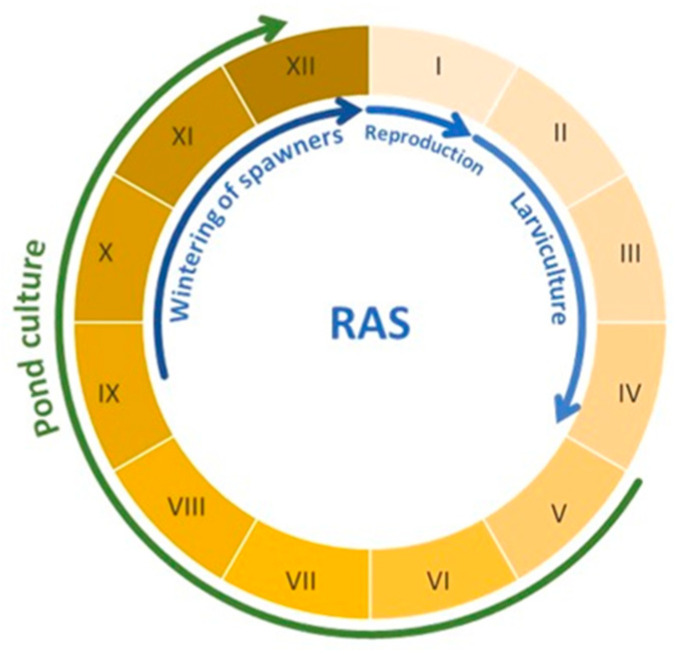
Preliminary assumptions of the Eurasian perch rearing cycle using the RAS and RAMPS. The pond rearing stage assumes both intensive production of perch and preparation of fish for sale (late autumn/winter period), during which the fish will not be fed. The letters I–XII indicate the numbers of the following months. Source: own elaboration.

**Table 1 animals-14-03100-t001:** Production assumptions for the calculation of production costs of Eurasian perch fry in the RAS.

Preparation of Spawners	Incubation of Eggs and Rearing of Larvae	Fry Rearing
Holding temperature: 6–12 °C Photoperiod: 9–13 h Hormonal injections: hcg dose (500 IU/kg) Manipulations: ms-222 solution (150 mg/L) Semen for fertilization: mixture from several males Feeding: three times a week, frozen ochre at a dose of 1% of fish biomass Daily water change: 5–10% of the total circulation Tank system: 3 × 1 m^3^	Temperature–incubation period of eggs: 8 days at 13 °C; in the next 2 days, increase to 17 °C Larvae rearing temperature: 17–21 °C Photoperiod: 24 h Larval rearing period: 45 days Daily water change: 5–10% of the total circulation Feeding: Artemia, commercial food Final weight: 0.5 g	Initial weight: 0.5 g Holding temperature: 21 °C Photoperiod: 24 h Daily water change: 10% of the total circulation Feeding: commercial food Final weight: 5 g

Source: own elaboration.

**Table 2 animals-14-03100-t002:** Direct operating costs of Eurasian perch fry production.

Cost Type	Unit	Expenditure	Unit Cost (PLN)	Cost (PLN)
Labor, including:	rgh	331	22.33	7391.23
Holding of spawners	rgh	112		
Incubation of eggs	rgh	174		
Fry rearing	rgh	45		
Electricity, including:	kWh	6676	0.63	4205.15
Holding of spawners	kWh	1958		
Incubation of eggs/initial rearing of larvae	kWh	2140		
Rearing of fry	kWh	2578		
Water consumption + sewage, of which:	m^3^	48.91	13.82	675.94
Holding of spawners	m^3^	22.50		
Incubation of eggs/initial rearing of larvae	m^3^	8.59		
Fry rearing	m^3^	17.82		
Chemicals, cleaning products, and others				500.00
Spawners manipulation				
MS-222	g	67.50	7.00	472.50
hCG	500 IU	3	5.00	15.00
Serra fluid:	pcs	2		160.29
Ethanol 70%	L	0.50	147.60	
Formaldehyde 40%	L	0.15	19.61	
Acetic Acid 99.5%	L	0.05	68.11	
Saline solution	L	0.50	11.88	5.94
Nutrition, including:				
Holding of spawners				
Ochotka	kg	1	56.00	56.00
Incubation of eggs/initial rearing of larvae				
Artemia	kg	3.11	507.00	1576.77
Perla 5	kg	8.52	71.75	611.31
Perla 4	kg	9.07	55.95	507.47
Rearing of fry				
Perla 4	kg	11.15	55.95	623.84
Nutra 1	kg	65.32	18.15	1185.56
Spawners	kg	3	25.00	75.00
Total costs of fry production (PLN)	18,062.00			

Source: own elaboration.

**Table 3 animals-14-03100-t003:** Share of particular cost components in direct operating costs of Eurasian perch fry production.

Cost Type	Cost (PLN)	Share (%)
Labor:	7391.23	41
Electricity:	4205.15	23
Water consumption + sewage, of which:	675.94	4
Chemicals, cleaning products, and others	500.00	3
Spawners manipulation	653.73	4
Nutrition	4635.95	25
Total costs of fry production (PLN)	18,062.00	100

Source: own elaboration.

**Table 4 animals-14-03100-t004:** Costs of Eurasian perch production in earthen ponds.

Itemization	Unit	Expen Diture	Unit Cost (PLN)	Cost (PLN)	Share (%)
Stocking material (fry, mean weight: 5 g)	pcs	29,334	0.50	14,667.00	25%
Feed	kg	3276	8.00	26,208.00	45%
Labor, including:	rgh	266	19.70	5240.00	9%
Stocking	rgh	16	19.70		-
Supervision	rgh	234	19.70		-
Catching	rgh	16	19.70		-
Electricity	kWh	9504	0.70	6653.00	11%
Depreciation (5%)	PLN	-	-	5682.00	10%
Total operating costs	-	-	-	58,450.00	100%

Source: own elaboration.

**Table 5 animals-14-03100-t005:** Investment expenditures of adaptation of one earthen pond with an area of 0.1 ha.

Itemization	Expenditure (PLN)	Share (%)
Vinyl cofferdam cross curtains	19,680.00	17
Inlet grating	6150.00	5
Inlet grating	6150.00	5
Longitudinal vinyl cofferdam curtains	51,000.00	45
Service footbridge	7380.00	6
Diffuser	12,300.00	11
Side blower	8600.00	8
Preparation of the pond on your own (5 days, three people 8 hrs each, PLN 19.7/hour)	2364.00	2
Total investment expenditures	113,624.00	100
Depreciation *	2841.00	-

* 5% per year, 50% per year equipment used for other purposes on the pond farm. Source: own elaboration.

## Data Availability

Data are included in the article. Additional experimental data are posted on the Pro-perch project website: https://pro-perch.infish.com.pl/ (accessed on 15 May 2024).

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
