# Peer review of "Optimizing Eurasian Perch Production: Innovative Aquaculture in Earthen Ponds Using RAS and RAMPS—Economic Perspective"

_animals, 2024, doi:10.3390/ani14213100_

Round 1
Reviewer 1 Report
Comments and Suggestions for Authors
The paper is interesting, but certain problems in analysis are noticeable.
Table 2 is hard to understand. For example, costs of water consumption are higher than costs of electricity, but water costs are not mentioned as important type of costs.
The same problem is recognized with costs of chemicals.
Therefore, table 2 should be divided in smaller tables (or presented in better way) in order to clarify the analysis.
Percentages discussed in the text below table 2 should be presented in table 2 (or in new - smaller tables).
- Costs of care are not mentioned in the Table 3? Are they indeed item named supervision in table 3? Costs of care (mentioned before table 3) = supervision (mentioned in table 3) = maintenance (mentioned after table 3)?
- Amount of depreciation is not the same in table 3 and 4?
- In row no. 322 the income was defined as the value of fish minus the cost of raising the fish. But it should be named profit instead of income.
- In row 326 – 327 authors used term revenue, without explaining the meaning of this term. Maybe revenue = value of the fish?
- Statement in row 334 (First sentence in section 3.4. Break even point – instead of term amount authors should use value of fish (value of production).
- In row 347 term amount of production is used to describe quantity of production.
Generally, terminology is not adequate and consistent!
For example – income = profit? Value of the fish = revenue = amount of production?
Author Response
The paper is interesting, but certain problems in analysis are noticeable.
Thank you for your detailed comments and suggestions. All of them have been included in the revised version of the manuscript and are marked with a red color.
Comments 1:
Table 2 is hard to understand. For example, costs of water consumption are higher than costs of electricity, but water costs are not mentioned as important type of costs.
The same problem is recognized with costs of chemicals.
Therefore, table 2 should be divided in smaller tables (or presented in better way) in order to clarify the analysis.
Percentages discussed in the text below table 2 should be presented in table 2 (or in new - smaller tables).
Response 1:
There were editing errors in Table 2. The data in the table have been reviewed and corrected. In order to improve the readability of the presentation of the results, a table has been added showing the share of certain cost components in the direct operating costs of Eurasian perch fry production. See Tables 2 and 3. Lines 309-330
-
Comments 2:
Costs of care are not mentioned in the Table 3? Are they indeed item named supervision in table 3? Costs of care (mentioned before table 3) = supervision (mentioned in table 3) = maintenance (mentioned after table 3)?
Response 2:
The costs of care, supervision, and maintenance were related to the same concept. Necessary corrections have been made in the text to make the results clearer. See Tables 2 and 3. Lines 309-330.
Comments 3:
- Amount of depreciation is not the same in table 3 and 4?
Response 3:
Table 4 (new 5) explains that 50% of the depreciation value was used in the calculations because 50% of the year the equipment is used for other purposes on the pond farm. See Table 5
Comments 4:
- In row no. 322 the income was defined as the value of fish minus the cost of raising the fish. But it should be named profit instead of income.
Response 4:
Profit is the difference between the amount of money spent and the amount of money earned in a given period, while income is the actual amount of money earned in a given period. Corrections have therefore been made in the text. See chapter 3.3
Comments 5:
- In row 326 – 327 authors used term revenue, without explaining the meaning of this term. Maybe revenue = value of the fish?
Response 5:
Revenue means the total amount of money generated by the activity over a given period, i.e. gross income before deducting any costs. Clarification added to the text. See lines 355-357
Comments 5:
- Statement in row 334 (First sentence in section 3.4. Break even point – instead of term amount authors should use value of fish (value of production).
Response 5:
BEP allows you to calculate the amount of production (not value) that guarantees cost recovery and no income. Lower production means losses, while higher production means profitability proportional to the increase in production. See chapter 3.4
Comments 6:
- In row 347 term amount of production is used to describe quantity of production.
Generally, terminology is not adequate and consistent!
For example – income = profit? Value of the fish = revenue = amount of production?
Response 6:
The text of the paper has been reviewed. To improve the readability and consistency of the text, errors in terminology were found and corrected. See chapter 3.2
Reviewer 2 Report
Comments and Suggestions for Authors
Dear Authors,
The paper offers an interesting economic perspective on the optimization of Eurasian perch production; however, several key areas require improvement.
The purpose of the paper is well formulated but could be presented more clearly in the abstract.
Regarding the methodology, the paper should explain certain economic assumptions, such as those related to market forecasts.
There is a lack of information on the variability of market prices, which could affect the final conclusions.
The paper lacks a clear articulation of the study's limitations and the research gap.
I suggest including future research directions, particularly in the context of further technological optimization.
Additionally, I recommend refining the structure of the conclusions so that they more clearly align with the presented research findings.
Best regards,
Author Response
The paper offers an interesting economic perspective on the optimization of Eurasian perch production; however, several key areas require improvement.
We thank the reviewer for his careful analysis of the manuscript and his suggestions for changes. All have been incorporated into the revised text. Changes in the text are highlighted in red.
Comments 1:
The purpose of the paper is well formulated but could be presented more clearly in the abstract.
Response 1:
Corrections have been made in the summary. Please see lines 27-31
Comments 2:
Regarding the methodology, the paper should explain certain economic assumptions, such as those related to market forecasts.
There is a lack of information on the variability of market prices, which could affect the final conclusions.
Response 2:
Changes have been made to both the Introduction and Discussion sections accordingly. See lines 131-142 and 513.
Comments 3:
The paper lacks a clear articulation of the study's limitations and the research gap.
Response 3:
The introduction chapter has been changed accordingly. See lines 251-258
Comments 4:
I suggest including future research directions, particularly in the context of further technological optimization.
Additionally, I recommend refining the structure of the conclusions so that they more clearly align with the presented research findings.
Response 4:
Suggested changes have been made both in the introduction and especially in the discussion sections, which have been restructured.
Reviewer 3 Report
Comments and Suggestions for Authors
Currently, the basis of pond farming in Eastern Europe is carp, and crucian carp is also grown (also for recreational purposes). Usually, commercial stocking of ponds with predatory fish species is not carried out, but they are specially stocked to suppress the number of weed fish and invasive species, for example, the sleeper, for ponds where a special water drain is not provided. The main limitations of the use of predatory fish species in aquaculture are the use of special feeds for the diet of predators, the specificity of pathogens characteristic of predatory species, as final hosts of parasites, such as helminths. The lack of proven methods for the prevention and treatment of pathogens, accumulation (accumulation) of pollutants, the manifestation of cannibalism - eating their own young, the presence of slow-growing groups that reach sexual maturity at small sizes, insufficient selection for selection by commercial and commercial characteristics, as well as the lack of economic justification for the effectiveness of artificial reproduction. All of the above mentioned factors prevent the spread of these species in aquaculture, etc. Thus, the article presented by the authors is relevant and answers a number of questions.
Note:
1. Lines 60-65: «The development of the farming of this group of fish is supported both by consumer expectations for healthy, high-quality animal protein and by the reduction of open water fishing, which is the main source of demand for products derived from these fish species. Currently, the culture of all perch fish in the world accounts for only about 1% of total aquaculture production, although new farms using recirculating aquaculture systems are steadily emerging in countries such as Switzerland, Ireland, France and Denmark.
Provide more detailed information on the schemes used to change aquaculture objects and justify the proposals of the authors of the study. Provide links to publications for the countries «Switzerland, Ireland, France and Denmark». It is necessary to add information on aquaculture in Italy.
2. Section «1.2. Percid fish aquaculture», lines 104-107: «Among predatory fishes, percid fishes (perch, pikeperch) are among the most highly 105 valued and achieve high market prices (in Poland and abroad). In recent years, these two 106 species have become the most frequently identified taxa for diversification of freshwater 107 aquaculture [20].»
It is necessary to provide additional information on the market of farmed freshwater fish species in the European EU countries, including taking into account the specifics of consumption, preferences, and freshwater aquaculture objects in different countries.
3. Lines 134-140: «Low or highly variable reproductive efficiency
• Low efficiency of larval rearing (lack of standardized procedures);
• Lack of rearing systems dedicated to perch;
• Lack of implementation of monosexual and triploid breeding as a result of genomic manipulation;
• Highly imperfect breeding programs that should ensure the maintenance of economically desirable traits over generations.» Among the problems of using perch fish in aquaculture, among the indicated problems are also: 1. the lack of measures tested in production for the prevention and treatment of diseases and monitoring of pathogens; 2. differences in growth rates, which is associated with the adaptation of species when living in the natural environment, including in conditions of freezes and resource limitations; 3. taking into account the factors causing the death of young fish, including larval aberrations and fungal diseases; 4. measures to compensate for damage and protect against predators, for example, cormorants Phalacrocorax carbo L. 1758, for the Masurian Lake District (Poland).
4. Reducing costs is also possible with the mass development of the concept of changing aquaculture objects, using a combination of pond farming with irrigation ("aquaponics") and other innovative approaches. In economic calculations, it is necessary to take into account the reduction of these costs, with an increase in the cost of electricity and wages. It is also necessary to add an innovative option related to the possibility of using alternative energy sources (since pond farms have the ability to accommodate generating facilities) and the use of water body cleaning (bottom sediments) for the production of organic fertilizers, and the water replaced from the ponds for irrigation of greenhouses, etc.
Author Response
Currently, the basis of pond farming in Eastern Europe is carp, and crucian carp is also grown (also for recreational purposes). Usually, commercial stocking of ponds with predatory fish species is not carried out, but they are specially stocked to suppress the number of weed fish and invasive species, for example, the sleeper, for ponds where a special water drain is not provided. The main limitations of the use of predatory fish species in aquaculture are the use of special feeds for the diet of predators, the specificity of pathogens characteristic of predatory species, as final hosts of parasites, such as helminths. The lack of proven methods for the prevention and treatment of pathogens, accumulation (accumulation) of pollutants, the manifestation of cannibalism - eating their own young, the presence of slow-growing groups that reach sexual maturity at small sizes, insufficient selection for selection by commercial and commercial characteristics, as well as the lack of economic justification for the effectiveness of artificial reproduction. All of the above mentioned factors prevent the spread of these species in aquaculture, etc. Thus, the article presented by the authors is relevant and answers a number of questions.
Thank you for your detailed review of the manuscript, your positive opinion, and your suggestions for changes. In the revised text of the manuscript, we have incorporated all of the reviewer's suggestions, which are highlighted in red.
Note:
Comments 1:
- Lines 60-65: «The development of the farming of this group of fish is supported both by consumer expectations for healthy, high-quality animal protein and by the reduction of open water fishing, which is the main source of demand for products derived from these fish species. Currently, the culture of all perch fish in the world accounts for only about 1% of total aquaculture production, although new farms using recirculating aquaculture systems are steadily emerging in countries such as Switzerland, Ireland, France and Denmark.
Provide more detailed information on the schemes used to change aquaculture objects and justify the proposals of the authors of the study. Provide links to publications for the countries «Switzerland, Ireland, France and Denmark». It is necessary to add information on aquaculture in Italy.
Response 1:
Suggested changes, including additional references, have been made in the Introduction chapter. See lines 108-120 and 128.
Comments 2:
- Section «1.2. Percid fish aquaculture», lines 104-107: «Among predatory fishes, percid fishes (perch, pikeperch) are among the most highly 105 valued and achieve high market prices (in Poland and abroad). In recent years, these two 106 species have become the most frequently identified taxa for diversification of freshwater 107 aquaculture [20].»
It is necessary to provide additional information on the market of farmed freshwater fish species in the European EU countries, including taking into account the specifics of consumption, preferences, and freshwater aquaculture objects in different countries.
Response 2:
Relevant information on consumption and aquaculture infrastructure has been added to the text of the manuscript. See lines 131-142
Comments 3:
- Lines 134-140: «Low or highly variable reproductive efficiency
- Low efficiency of larval rearing (lack of standardized procedures);
- Lack of rearing systems dedicated to perch;
- Lack of implementation of monosexual and triploid breeding as a result of genomic manipulation;
- Highly imperfect breeding programs that should ensure the maintenance of economically desirable traits over generations.» Among the problems of using perch fish in aquaculture, among the indicated problems are also: 1. the lack of measures tested in production for the prevention and treatment of diseases and monitoring of pathogens; 2. differences in growth rates, which is associated with the adaptation of species when living in the natural environment, including in conditions of freezes and resource limitations; 3. taking into account the factors causing the death of young fish, including larval aberrations and fungal diseases; 4. measures to compensate for damage and protect against predators, for example, cormorants Phalacrocorax carbo L. 1758, for the Masurian Lake District (Poland).
Response 3:
Thank you for pointing out the potential problems with perch rearing. This information has been added to the text of the paper. See lines 131-142
Comments 4:
- Reducing costs is also possible with the mass development of the concept of changing aquaculture objects, using a combination of pond farming with irrigation ("aquaponics") and other innovative approaches. In economic calculations, it is necessary to take into account the reduction of these costs, with an increase in the cost of electricity and wages. It is also necessary to add an innovative option related to the possibility of using alternative energy sources (since pond farms have the ability to accommodate generating facilities) and the use of water body cleaning (bottom sediments) for the production of organic fertilizers, and the water replaced from the ponds for irrigation of greenhouses, etc.
Response 5:
Thank you for pointing this out, this is exactly what we had in mind when we wrote about product or infrastructure innovation in the discussion chapter. More information is included in the discussion chapter. See lines 498-501 and 522-524.
Round 2
Reviewer 1 Report
Comments and Suggestions for Authors
Authors have made necessary corrections concerning errors in table 2 and errors in terminology.
There is only one additional correction that should be made – line 370 – 371.
Instead of: “The break-even point refers to the volume of production that is equal to the costs incurred” there should be written: “The break-even point refers to the volume of production that generates revenues equal to the costs incurred”.
Author Response
Comments 1:
Authors have made necessary corrections concerning errors in table 2 and errors in terminology.
There is only one additional correction that should be made – line 370 – 371.
Instead of: “The break-even point refers to the volume of production that is equal to the costs incurred” there should be written: “The break-even point refers to the volume of production that generates revenues equal to the costs incurred”.
Response 1:
Thank you again for your guidance and help in improving the manuscript. The suggested change has been made to the text at the indicated location in the manuscript. Changes are highlighted in red and blue.
Reviewer 3 Report
Comments and Suggestions for Authors
The authors have made a number of changes and improved the publication. Please note the following issues for which there is insufficient information or no references:
General comments:
1. Currently, the technology for monitoring the cultivation of river perch in various conditions is not sufficiently developed. In addition to economic assessments, a protocol for analyzing the condition of the organism of this aquaculture object and a checklist of possible pathogens and threats during cultivation are also required. There is insufficient published data on this issue (Keslemont et al., 2001, DOI: 10.1016/S0044-8486(01)00615-9).
2. According to many authors (Benrmann-Godel et al., 2015, DOI: 10.1201/b18806-9), river perch has been shown to have high resistance to various abiotic and biotic environmental factors, including parasitic invasion. However, active introduction into aquaculture, maintenance at high density and selection may reduce resistance to parasitic infestation and require additional preventive measures and costs.
Other comments:
3. Line 29: "Eurasian perch (Perca fluviatilis L.)", species name in Latin in italics.
Author Response
Comments 1:
The authors have made a number of changes and improved the publication. Please note the following issues for which there is insufficient information or no references:
Response 1:
Thank you again for your tips on how to improve the manuscript. All suggestions have been incorporated into the revised version of the manuscript. Changes are highlighted in blue.
General comments:
Comments 2:
- Currently, the technology for monitoring the cultivation of river perch in various conditions is not sufficiently developed. In addition to economic assessments, a protocol for analyzing the condition of the organism of this aquaculture object and a checklist of possible pathogens and threats during cultivation are also required. There is insufficient published data on this issue (Keslemont et al., 2001, DOI: 10.1016/S0044-8486(01)00615-9).
Response 2:
Thanks for this valuable tip. Citation suggestions have been added to the Introduction. Suggested content and citation have also been added to the Discussion.
Comments 3:
- According to many authors (Benrmann-Godel et al., 2015, DOI: 10.1201/b18806-9), river perch has been shown to have high resistance to various abiotic and biotic environmental factors, including parasitic invasion. However, active introduction into aquaculture, maintenance at high density and selection may reduce resistance to parasitic infestation and require additional preventive measures and costs.
Response 3:
Thank you for your comments. This aspect of perch rearing is very important, and issues related to this topic have been introduced in the previous revised version of the manuscript and additionally in the current version. In our opinion, this aspect is now sufficiently supported in the current version of the manuscript by references that address this issue.
Other comments:
Comments 4:
- Line 29: "Eurasian perch (Perca fluviatilis L.)", species name in Latin in italics.
Response 4:
The suggested change has been implemented.